# A General Stitching Solution for Whole-Brain 3D Nuclei Instance Segmentation from Microscopy Images

**Ziquan Wei**[1]                                                ZIQUANW@EMAIL.UNC.EDU
[1] *Department of Psychiatry, University of North Carolina at Chapel Hill*
**Guorong Wu**[1,2]                                             GUORONG_WU@MED.UNC.EDU
[2] *Department of Computer Science, University of North Carolina at Chapel Hill*

## Abstract

High-throughput 3D nuclei instance segmentation (NIS) is critical to understanding the complex structure and function of individual cells and their interactions within the larger tissue environment in the brain. Despite the significant progress in achieving accurate NIS within small image stacks using cutting-edge machine learning techniques, there has been a lack of effort to extend this approach towards whole-brain NIS. To address this challenge, we propose an efficient deep stitching neural network built upon a knowledge graph model characterizing 3D contextual relationships between nuclei. Our deep stitching model is designed to be agnostic, enabling existing limited methods (optimized for image stack only) to overcome the challenges of whole-brain NIS, particularly in addressing the issue of inter- and intra-slice gaps. We have evaluated the NIS accuracy on top of state-of-the-art deep models, such as Cellpose, with $128 \times 128 \times 64$ image stacks.

**Keywords:** Image stitching, 3D microscopy image, Whole-brain nucleus instance segmentation, Graph neural network.

## 1. Introduction

Light-sheet microscopy is a powerful imaging modality that allows for fast and high-resolution imaging of large samples, such as the whole brain of the mouse (Yang et al., 2022; Bennett and Kim, 2022). Alternatively, tissue-clearing techniques enable the removal of light-scattering molecules, thus improving the penetration of light through biological samples and allowing for better visualization of internal structures, including nuclei (Banerjee and Poddar, 2022; You et al., 2023). Together, light-sheet microscopy and tissue-clearing techniques have revolutionized the field of biomedical imaging and they have been widely used for studying the structure and function of tissues and organs at the cellular level.

Accurate 3D nuclei instance segmentation plays a crucial role in identifying and delineating individual nuclei within three-dimensional space, which is essential for understanding the complex structure and function of biological tissues in the brain. However, due to the high cost of 3D manual nuclei annotations and the complexity of learning, current end-to-end NIS models are typically limited to training and testing on small image stacks (e.g., $128 \times 128 \times 64$). Considering these limitations, one approach for achieving whole-brain NIS is dividing the whole stack into smaller stacks so that the existing NIS methods can handle each piece individually. In such a scenario, constructing the whole-brain nuclei instance segmentation in 3D from these smaller image stacks arises a new challenge. The gaps between these smaller stacks (intra-slice) and the slices (inter-slice) require a robust stitching method for accurate NIS. Although Cellpose offers a straightforward solution for 3D input,

the extremely high RAM demand of a whole-brain image of a P4 mouse, which can be as much as 3TB in theory, necessitates the use of the dividing and stitching method.

To address these gap issues, We propose a stitching framework for whole-brain NIS as shown in Figure 1. We evaluated this hierarchical stitching framework by the segmentation precision and stitching accuracy (correspondence matching between 2D masks). Compared to no stitching, Our deep stitching model has shown a significant improvement in the NIS results with different state-of-the-art models, and indicating its potential for practical applications in the field of neuroscience.

## 2. Method and Experiment

In the hierarchical stitching framework, there are two stages:

1. *Resolve intra-slice gap in X-Y plane.* Suppose that each within-stack NIS result overlaps with its neighboring image stack in the X-Y plane. By doing so, each 2D nuclei instance in the X-Y plane is expected to have at least one "gap-free" NIS estimation that does not touch the boundary of the image stack. Thus, we can resolve the intra-slice gap problem in X-Y plane in three steps: (i) identify the duplicated 2D nuclei instances from multiple overlapped image stacks, (ii) find the representative NIS result from the "gap-free" image stack, and (iii) unify multiple NIS estimations by using the "gap-free" NIS estimation as the appearance of the underlying 2D nuclei.

2. *Inter-slice stitching using graph contextual model.* At each gap area along Z-axis, we deploy the graph contextual model, which has two MLP components, to stitch the sliced nuclei instances. Specifically, we follow the partition of the whole-brain microscopy image in stage 1, that is a set of overlapped 3D image stacks, all the way from the top to the bottom as shown in the bottom-right corner of Figure 1. It is worth noting that each 2D

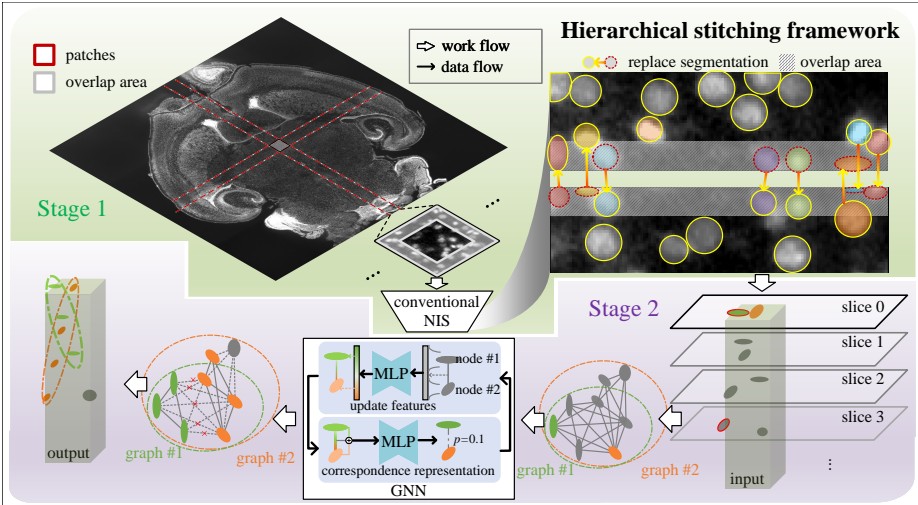

Figure 1: The proposed hierarchical stitching framework for whole-brain NIS. Top: Resolve the intra-slice gap in the X-Y plane by overlap. Bottom: Graph contextual model for inter-slice gap.

nuclei instance in the X-Y plane is complete, as indicated by the red-highlighted portion extending beyond the image stack. Next, we assign a stack-specific local index to each 2D nuclei instance. After that, we apply the (trained) graph contextual model to each 2D nuclei instance by (i) constructing the contextual graph centered at the underlying 2D nuclei instance, (ii) predicting the spatial correspondences with respect to the neighboring 2D instances.

**Graph contextual model.** *First*, we construct an initial contextual graph $G = \{\mathbf{V}, \mathbf{E}\}$ for each 2D nucleus instance $x$ (i.e., image appearance vector, gradient flow by Cellpose (Stringer et al., 2021; Pachitariu and Stringer, 2022) The set of nodes $\mathbf{V} = \{x_i | \mathcal{D}(x, x_i) > \delta\}$ includes all neighboring 2D nuclei instances, where the distance between two is denoted by $\mathcal{D}$, and $\delta$ is a threshold. The matrix $\mathbf{E} \in \mathbb{R}^{N \times N}$ represents the edges between nodes. *Second,* we train the model on a set of contextual graphs $G$ to recursively

1. *Graph feature representation learning.* For the $k^{th}$ iteration, we enable two connected nodes to exchange their feature representations constrained by the current relationship topology $e_{ij}^k$ by the $k^{th}$ layer of the deep stitching model. In this context, we define the message-passing function as $x_i^{(k+1)} = \gamma^{(k+1)} \left( x_i^{(k)}, \Sigma_{j \in \mathcal{N}(i)} \phi^{(k)}(x_i^{(k)}, x_j^{(k)}, e_{j,i}^{(k)}) \right)$. Following the popular learning scheme in knowledge graphs (Wang et al., 2021), we employ Multilayer Perceptron (MLP) to act functions $\gamma, \phi$.

2. *Learning the link-wise similarity function to predict nuclei-to-nuclei correspondence.* Given the updated node feature representations $\{x_i^{(k+1)}\}$, we train another MLP to learn the similarity function $\psi$ in a layer-by-layer manner. In the $k^{th}$ layer, we update each 2D-to-3D contextual correspondence $e_{j,i}^{(k+1)}$ for the next layer by $e_{j,i}^{(k+1)} = \psi^{(k+1)} \left( x_i^{(k+1)}, x_j^{(k+1)}, e_{j,i}^{(k)} \right)$.

**Data and computing environment.** In the following experiments, we have trained Mask-RCNN-R50, Mask-RCNN-R101, and CellPose on X-Y plane of 16 image stacks ($128 \times 128 \times 64$), which include in total 6,847 manually labeled 3D nuclei.

**Quantitative evaluation.** As shown in Figure 2, there is a clear sign that NIS models with our hierarchical stitching method outperform IoU-based counterparts on NIS metrics, regardless of the NIS backbone models. In average, our hierarchical stitching method has improved 14.0%, 5.1%, 10.2%, and 3.4% in precision, recall, F1 score, and stitching accuracy, respectively compared with IoU-based no stitching results.

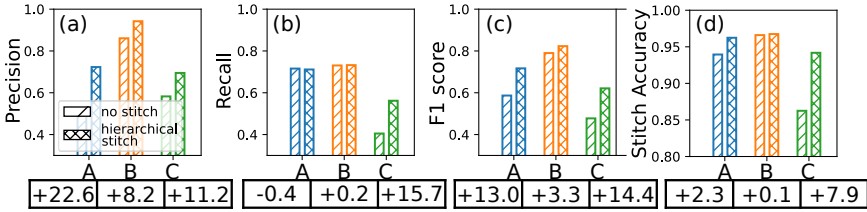

Figure 2: The NIS precision (a), recall (b), F1 score (c), and stitching accuracy (d) by stitching or not, where the NIS backbones include Mask-RCNN-R50 (A, blue), Mask-RCNN-R101 (B, orange), CellPose (C, green).

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
