# OpenReview forum: "A General Stitching Solution for Whole-Brain 3D Nuclei Instance Segmentation from Microscopy Images"
_MIDL.io/2023/Short_Paper_Track — MIDL 2023 Short paper track Poster_

### Official Review · Reviewer_k6QM · 2023-04-24

**Rating:** 7
**Confidence:** 5

**Review:**

This paper proposed to leverage a graph model to characterize contextual relationships in 3D. This improves over existing stitching solutions, which are mainly focused on 2D approaches.

- The idea of extending 2D approaches to a 3D context employing a graph model is interesting.

- Improvements of the proposed approach compared to the non-hierarchical version are consistent across the different metrics.

- Maybe discussing and comparing to other existing 3D approaches could have improved the abstract. It was unclear whether this is the first attempt for doing 3D instance segmentation, or just a better alternative to existing approaches.

---

### Official Review · Reviewer_PKjR · 2023-04-26
**An interesting stitching method for Whole-Brain 3D Nuclei Instance**

**Rating:** 7
**Confidence:** 4

**Review:**


This paper addresses 3D nuclei instance segmentation (NIS). While most of the existing literature focused on NIS on small image stacks, addressing the problem on
whole-brain NIS remains less investigated, despite its importance.

The authors proposed a deep stitching neural network built upon a knowledge graph characterizing 3D contextual relationships between nuclei. Specifically, they deploy a graph contextual model at each gap area along the Z-axis, with two MLP components, enabling stitching the sliced nuclei instances. While current methods are optimized for and limited to image stack only, the proposed solutions mitigates the difficulties of whole-brain NIS, specifically the issues of inter- and intra-slice gaps. The experiments showed significant improvements (in precision, recall, F1 score, and stitching accuracy), in comparison to the results obtained without stitching.

Overall, this is a well-executed paper. It is clear and well-written, with each component being well described. Also, the results are convincing. Therefore, I recommend acceptance.